# SpintBench: Evaluating LLMs' Complex Reasoning via Spatial Integration Challenges

## Abstract

Large language models (LLMs) have demonstrated remarkable reasoning capabilities across diverse domains, yet their comprehensive spatial reasoning competencies remain underexplored. This paper proposes a benchmark construction framework for evaluating spatial reasoning in both 2D and 3D spaces—one that requires LLMs to infer global information from provided local details through spatial integration. Specifically, we have designed rules to automatically generate spatial descriptions of local scenes with overlapping cues, as well as corresponding question-answer (QA) pairs, forming the spatial integration reasoning benchmark SpintBench. Experimental results show that state-of-the-art (SOTA) LLMs still struggle to tackle **SpintBench** effectively: while the combination of few-shot learning and chain-of-thought (CoT) prompting yields modest performance improvements, these gains remain limited. This work is expected to provide valuable insights for advancing the investigation of spatial reasoning capabilities in LLMs.

## 1 Introduction

Spatial reasoning is an important cognitive process for both humans and other animals, which is grounded in mental representations of spatial objects as well as relationships between them, and influences interactions with physical space (Park et al., 2020; Ho et al., 2022; Peer & Epstein, 2021). Therefore, it is essential to enhance LLMs' spatial reasoning so as to improve their comprehension of the environment. Currently, though LLMs still face great challenges in reasoning, more o1-like thinking models emerge and are characterized by slow-thinking, which place a higher demand for evaluation, since more distinctive and challenging evaluation datasets would be effective incentives for the promotion of LLMs.

Previous studies have identified limitations of Large Language Models (LLMs) in multi-hop reasoning, encompassing logical reasoning (Patel et al., 2024; Yang et al., 2025) and advanced theory of mind (Wu et al., 2023), among other domains. Nevertheless, such multi-hop challenges have rarely been incorporated into the design of spatial reasoning tasks—particularly in spatial integration scenarios—resulting in existing spatial benchmarks that are either overly simplistic or lack discriminatory power.

To bridge this gap, we propose a novel dataset, Spatial Integration Benchmark (SpintBench). Inspired by the **transitive inference**—a paradigm that assesses reasoners' capacity to infer "A->C" from the given premises "A->B" and "B->C"—we extend this framework from one-dimensional (1D) to two-dimensional (2D) space (Fig.1). In this extended context, the spatial integration task necessitates that reasoners synthesize information across local spaces using contextual cues and derive insights that depend on global-level information acquisition.

Our contributions can be summarized as follows. First, we proposed the first spatial reasoning benchmark which combines multi-hop reasoning with spatial integration in both 2D and 3D space, improving the difficulty of tasks to a large extent. Besides, the benchmark construction method we designed boasts the advantages of automation and high scalability, while also being resistant to data contamination.

Second, this study systematically evaluates the performance of SOTA models on **SpintBench**, revealing the shortcomings of current models in complex spatial reasoning. The evaluation results also indicate that **SpintBench** exhibits both high difficulty and strong discriminative power.

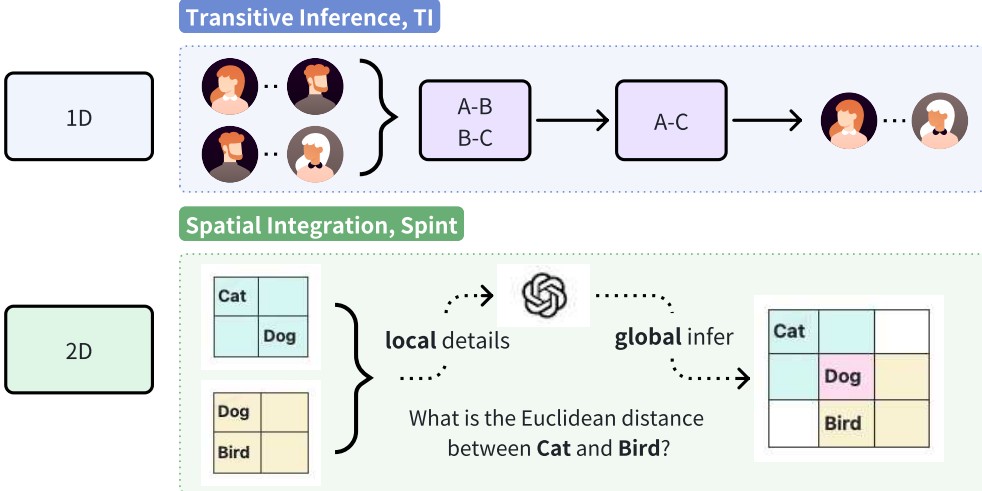

Figure 1: Transitive inference which requires induction "A→C" from the given premises "A→B" and "B→C" is extended to 2D spatial integration. **SpintBench** largely enhances the difficulty of spatial reasoning.

Third, through comprehensive experiments, this study explores the detailed impacts of spatial parameters related to spatial integration tasks on task difficulty and model performance. It reveals that, within a reasonable range, the sparsity of objects in space affects task difficulty—providing the applicable scope and difficulty control methods for the reasoning framework proposed in this study.

Fourth, attempts to improve model performance are made via prompting experiments, which show that the combination of Chain-of-Thought (CoT) and few-shot learning contributes to enhancing model performance to a certain extent.

## 2  RELATED WORK

### 2.1  EVALUATION OF SPATIAL REASONING IN LLMS

Spatial reasoning in LLMs has not yet been thoroughly covered by comprehensive evaluation. Existing works mostly focus on multi-modal QA benchmarks, such as GQA (Hudson & Manning, 2019) and NLVR (Suhr et al., 2017), which express spatial phenomena through visual images. However, spatial reasoning in natural language is also very important for LLMs involved in many natural language understanding (NLU) tasks. In this respect, bAbI is the first positional reasoning dataset providing direct textual spatial question answering (TQA, Task 17) (Weston et al., 2015) though both its scenes and questions are rather simplified. SpartQA is another TQA benchmark for spatial reasoning, whose questions require more deeper reasoning and encompass four types (find relation, find blocks, choose object, and yes/no) (Mirzaee et al., 2021). Findings based on StepGame Benchmark reveal the limitations of LLMs in multi-hop reasoning (Shi et al., 2022; Li et al., 2024), and the room for improvement still remains (Yamada et al., 2024).

A recent study subdivides spatial intelligence into six fundamental capabilities: metric measurement, mental reconstruction, spatial relations, perspective-taking, deformation and assembly, and comprehensive reasoning (Cai et al., 2025). However, atomic spatial intelligence across different dimensions may coexist in the same task, which assesses the comprehensive performance of a model's multi-dimensional spatial intelligence. **SpintBench**, proposed in this paper, integrates several atomic spatial abilities such as metric measurement, mental reconstruction, comprehensive reasoning within a single task.

## 2.2 Transitive inference and Spatial Integration

Transitive inference (TI) stands as the most canonical task paradigm for examining relational cognition, which necessitates that subjects derive the conclusion A > C by leveraging the premises A > B and B > C. The transitivity denoted by ">" involves synthesizing local premise components into an ordered sequence, with the letters A, B, and C representing entities from any given domain (Wu & Deng, 2025). This type of relational cognition bears certain similarities to traditional assessments of spatial relations, yet it is not entirely identical. Transitive inference has a broader scope, not limited to spatial relations, and its inferential chain can be extended to a considerable length. In contrast, traditional evaluations of spatial relations mostly focus on low-order (first or second order) reasoning. Therefore, migrating the approach of assessing 1D reasoning through transitive inference to spatial reasoning evaluation represents a promising exploration. Specifically, this approach can be extended to 2D and 3D spaces, where the reasoner needs to integrate local spatial clues to infer global spatial information, namely **spatial integration**.

## 2.3 Benefits of prompting and multi-agent system

Prompt engineering shows promise in promoting LLMs' spatial reasoning performance (Sharma, 2023), since several prompting methods have been proposed to be effective, such as Incontext Learning (ICL) (Brown et al., 2020), Chain-of-Thought (CoT)(Wei et al., 2022), ReAct (Yao et al., 2023b), and Tree-of-Thought (ToT) (Yao et al., 2023a). Multi-agent systems have also been proven to achieve better results than single agents in solving complex problems through the division of labor and collaboration among multiple sub-agents.

## 3 SpintBench Construction

Drawing inspiration from transitive inference, we propose an automated approach to generating spatial reasoning tasks via inverse construction. Specifically, we first generate a global map containing randomly distributed objects, then derive local spaces by splitting this global map—with adjacent local spaces featuring overlaps to provide cues for reasoning and spatial integration. This generation method is applicable to both 2D and 3D spaces, thus offering good extensibility. The following sections will elaborate on the construction process in the 2D scenario, which can be divided into the following steps.

First, **global space generation and object distribution**. A global space is constructed as a square grid with side length $m$, comprising $m*m$ cells where each cell measures 1 unit in length (Fig. 1). Subsequently, $n$ distinct objects are randomly positioned within this generated global space, with each object occupying a unique cell in the $m*m$ grid. These objects are distributed in a manner that ensures they reside in separate cells.

Second, **local space partition**. The global space embedded with objects is subdivided into local spaces, each configured as a square grid with side length $k$ (where $k<m$) and containing $k*k$ cells. Adjacent local spaces overlap by one row or one column, with the overlapping objects acting as cues for inference. To ensure adequate coverage of the entire global space, $m$ and $k$ must satisfy the following equality relationship: $m - k = z(k - 1), \ z \in \mathbb{Z}$, so as to guarantee the structured overlapping of local spaces.

Finally, **problem generation and filtration**. The two steps mentioned above lay the foundation for question generation in this step. Specifically, the position information of objects within local spaces is provided as the context story, which can be either in an *ordered* form—where the descriptions of local grids follow their sequence (Fig.2) —or in a *shuffled* form, where the descriptions are based on randomly disordered local grids. Theoretically, processing a shuffled context story should be more challenging than handling an ordered one, as the latter allows for easier utilization of cues found in the overlapping regions of adjacent local spaces.

Then a problem prompt is generated by randomly selecting two objects from the $n$ objects. LLMs are tasked with inferring the Euclidean distance between the two selected objects. The total number of possible problem prompts is $C_n^2 = \frac{n(n-1)}{2}$. However, not all generated prompts are included in **SpintBench**; only those that require models to **infer** rather than merely recall are incorporated. Thus, the prompts generated above undergo filtration to form **SpintBench**. Additionally, to investigate

how LLMs perform differently under these varying conditions—namely recall versus inference, and ordered versus shuffled—we have conducted experiments, the details of which are elaborated in the following section.

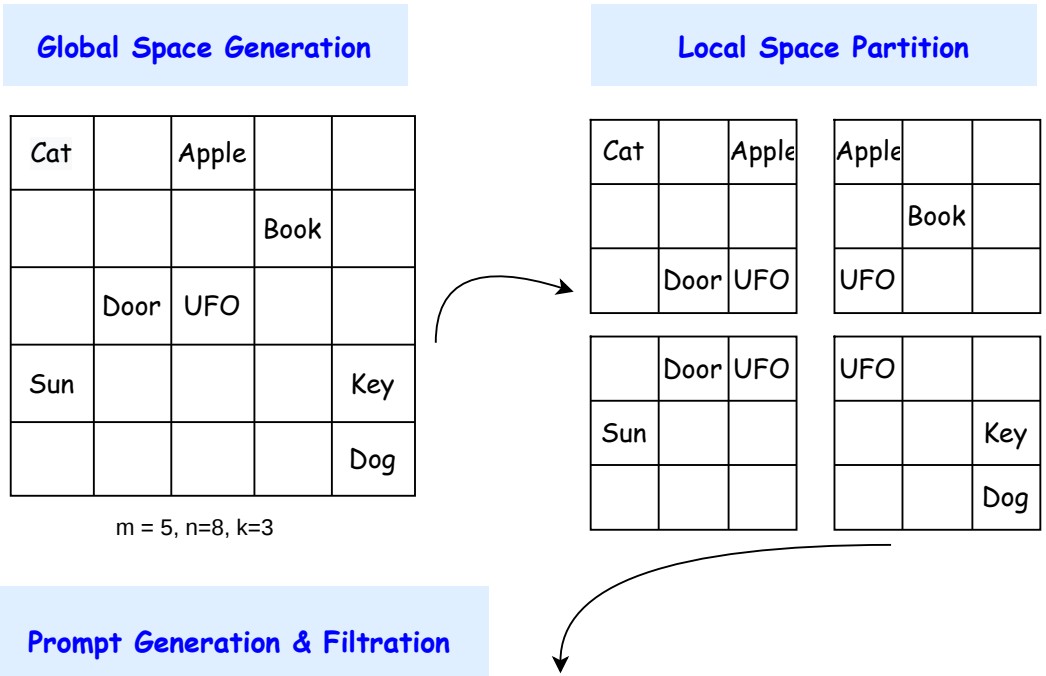

**Global Space Generation**

**Local Space Partition**

m = 5, n=8, k=3

**Prompt Generation & Filtration**

**STORY (ordered)**: Suppose there are **8** objects on a **5\*5** 2D plane, and each object is located in a small square cell. The length of each cell is 1 unit. Each time you can only see the plane of grid size **3\*3**. Now you see four 3\*3 grids. The relative coordinates of the objects inside are as follows:
**Local grid 1**: Cat (local coordinates(1, 1)), Apple (local coordinates(3, 1)), Door (local coordinates(2, 3)), UFO (local coordinates(3, 3)); **Local grid 2**: Apple (local coordinates(1, 1)), UFO (local coordinates(1, 3)), Book (local coordinates(2, 2)); **Local grid 3**: Door (local coordinates (2, 1)), UFO (local coordinates (3, 1)), Sun (local coordinates (1, 2)); **Local grid 4**: UFO (local coordinates (1, 1)), Key (local coordinates (3, 2)), Dog (local coordinates (3, 3)).

**QUESTION (infer)**: Please infer what the Euclidean distance is between **Cat** and **Dog**? Please keep two decimal places for the answer.
**ANSWER**: 7.07

Figure 2: The construction process of **SpintBench**. The dataset construction involves three core steps: (1) Global space generation and object distribution, where a square grid is created and objects are randomly placed in distinct cells; (2) Local space partitioning, subdividing the global space into overlapping local grids to ensure coverage; and (3) Problem generation and filtration, where object position information is used to create ordered or shuffled context stories for question generation, with shuffled stories posing a greater challenge.

## 4 EXPERIMENTS

### 4.1 EXPERIMENT DESIGN

**Experimental Conditions**. Given that the construction of the global space and the partitioning of local spaces are constrained by several parameters—specifically, $m$ (the side length of the global space), $n$ (the number of objects), and $k$ (the side length of the local space)—we selected the parameter combination **[m, n, k] = [9, 25, 3]** in 2D space to examine how LLMs perform differently under the recall versus inference conditions, as well as under ordered versus shuffled context stories provided to the models. This results in four data groups: *recall\*ordered*, *recall\*shuffled*, *infer\*ordered*, and

*infer\*shuffled*. For the *infer* conditions, one hundred data samples were randomly selected as the test set. For the *recall* conditions, by contrast, all available data were included in the test set, as such data are relatively scarce and only exist within local spaces. Furthermore, we extended the identical construction approach to **3D space** to investigate whether comparable effects persist. For the 3D space setup, we selected the parameter combination **[m, n, k] = [9, 80, 3]**; specifically, the number of objects was set to 80, ensuring the object density within the space remains at a suitable level.

**Parameter Experiment**. To ensure that the findings from the aforementioned experiments are not sensitive to specific parameter values, we further examine the respective influences of the global space size (*m*) and the number of objects (*n*) by varying their values (see Table.1).

**Prompting Engineering Experiment**. Considering that SOTA models exhibit poor performance in the *infer-shuffled* task condition under zero-shot setting (see detailed reports in subsequent sections), which indicates that models have relatively weak spatial integration and reasoning abilities, we further explore two prompting strategies that may effectively enhance LLMs' reasoning performance, namely **one-shot CoT(chain-of-thought)** and **one-shot ReAct**. The corresponding prompt templates are available in the Appendix. Notably, this set of prompting experiments is exclusively conducted under the most challenging *infer-shuffled* condition, with the parameter configuration specified as *[m, n, k] = [9, 25, 3]*.

## 4.2 EVALUATION SETTINGS

A total of 17 LLMs from seven leading manufacturers are evaluated under four experimental conditions, all accessed uniformly via their respective APIs. These include OpenAI's models—OpenAI-o3-high.code, OpenAI-o4-mini.high.0416.code, along with non-thinking models like OpenAI-gpt4.1-0414 and GPT4o-1120; Anthropic's Claude-4-Sonnet-thinking-gcp and Claude-4-Opus-thinking-gcp; Google's Gemini-2.5-Flash and Gemini-2.5-Pro; Qwen's QwenAPI-3-235b-a22b-thinking, QwenAPI-qwq-plus.latest, QwenAPI-3-32b, and QwenAPI-max-latest; DeepSeek's DeepSeekAPI-R1-0528.volc.forCompetitor and DeepSeekAPI-V3-0324.volc.forCompetitor; Seed's thinking models namely Doubao-1.6-thinking.0715.foreval and Doubao1.5-pro-32k.250115; and Moonshot's reasoning model Kimi-K2.volc.foreval.

In the investigations into the influence of parameter values and the extension to 3D space, only four models were selected for evaluation to reduce costs. These include two thinking models (DeepSeek-R1, Doubao1.6-thinking-0715) and two non-thinking models (GPT-4o, Doubao1.5-pro-32k), with a relatively distinct performance gap between the two categories.

**Evaluation metric**. Since the tasks in these experiments require models to predict the Euclidean distance between two given objects, a model's prediction is considered correct if it falls within a small range around the ground truth (with an error margin of 0.01). This narrow range is sufficient to avoid incorrect judgments.

## 5 RESULTS AND DISCUSSIONS

### 5.1 SHORTAGE IN LLMS' INFERENCE AND ORDER EFFECT

**The condition of *infer* is much harder than that of *recall***. Results across different experimental conditions show that (Table.2), most SOTA thinking models achieve nearly full marks in the *recall* condition, whereas their performance decreases largely when it comes to the *infer* condition. Across all models, average score under *recall* is near 90%, while that under *infer* is near 50% and even lower.

**Order effects are evident in the reasoning performance of LLMs**. In the *infer* condition, all models perform significantly better when provided with an ordered context story compared to a shuffled one. In the *recall* condition, SOTA thinking models such as OpenAI-o4-mini show little difference between ordered and shuffled context stories, as their scores in both scenarios approach the ceiling. However, for weaker models that struggle most in the *infer* condition—such as OpenAI-gpt4.1-0414—performance is relatively stronger when given a shuffled context story. This phenomenon may arise because weaker models are likely confused by overlapping areas between adjacent local spaces, rather than leveraging these overlaps as cues to support effective reasoning.

Table 1: Model performance across four conditions in 2D space (%). SpintBench refers to the infer-shuffled column. $\Delta$ denotes the difference between shuffled and ordered scores (shuffled - ordered).

| Model | Infer | | | Recall | | | avg_score |
|---|---|---|---|---|---|---|---|
| | shuffled | ordered | $\Delta$ | shuffled | ordered | $\Delta$ | |
| OpenAI-o3-high.code | 67.00 | 88.00 | -21.00 | 100.00 | 100.00 | 0.00 | 88.75 |
| Gemini-2.5-Flash | 52.00 | 85.00 | -33.00 | 100.00 | 100.00 | 0.00 | 84.25 |
| Gemini-2.5-Pro | 44.00 | 92.00 | -48.00 | 96.08 | 98.04 | -1.96 | 82.53 |
| OpenAI-o4-mini.high.0416.code | 39.00 | 95.00 | -56.00 | 98.04 | 98.04 | 0.00 | 82.52 |
| Claude-4-Sonnet-thinking | 36.00 | 87.00 | -51.00 | 100.00 | 100.00 | 0.00 | 80.75 |
| Claude-4-Opus-thinking | 33.00 | 95.00 | -62.00 | 98.04 | 100.00 | -1.96 | 81.51 |
| DeepSeekAPI-R1-0528 | 33.00 | 68.00 | -35.00 | 100.00 | 100.00 | 0.00 | 75.25 |
| Doubao-1.6-thinking.0715 | 33.00 | 69.00 | -36.00 | 96.08 | 96.08 | 0.00 | 73.54 |
| QwenAPI-3-235b-a22b-thinking | 31.00 | 83.00 | -52.00 | 100.00 | 100.00 | 0.00 | 78.50 |
| QwenAPI-qwq-plus.latest | 18.00 | 45.00 | -27.00 | 84.31 | 84.31 | 0.00 | 57.91 |
| Kimi-K2.volc.foreval | 8.00 | 32.00 | -24.00 | 96.08 | 96.08 | 0.00 | 58.04 |
| DeepSeekAPI-V3-0324 | 8.00 | 15.00 | -7.00 | 84.31 | 82.35 | 1.96 | 47.42 |
| QwenAPI-3-32b | 8.00 | 40.00 | -32.00 | 70.59 | 62.75 | 7.84 | 45.34 |
| QwenAPI-max-latest | 2.00 | 7.00 | -5.00 | 62.75 | 54.90 | 7.85 | 31.66 |
| OpenAI-gpt4.1-0414 | 1.00 | 11.00 | -10.00 | 94.12 | 90.20 | 3.92 | 49.08 |
| Doubao1.5-pro-32k.250115 | 1.00 | 3.00 | -2.00 | 74.51 | 78.43 | -3.92 | 39.24 |
| GPT4o-1120 | 1.00 | 4.00 | -3.00 | 41.18 | 37.25 | 3.93 | 20.86 |

**SpintBench poses significant challenges even for SOTA thinking models**. Focusing on the *infer\*shuffled* condition— the sole scenario included in SpintBench—results indicate that non-thinking models typically achieve very low scores, with some (such as GPT-4o-1120) even approaching zero. Among SOTA thinking models, OpenAI o3-high-code, despite being the top performer, reaches only 67%, while all other thinking models also struggle. Furthermore, SpintBench is distinctive in its ability to reveal substantial performance gaps between different models, underscoring both its difficulty and its value as a benchmark.models, highlighting both its difficulty and its value as a benchmark.

**Effects keep consistent in 3D space**. Results in 3D space show a pattern similar to that in 2D space (Table.2). Despite slight differences in the models' partial orders, Gemini-2.5 and OpenAI-o3-high.code still belong to the first tier. In addition, compared with the *recall* task, all models perform significantly worse in the *infer* task, indicating that LLMs' ability to perform reasoning across local spaces is relatively limited. Additionally, in the *infer* task, models perform worse under the *shuffled* context story condition than under the *ordered* condition, especially for thinking models, since the scores of non-thinking models are close to zero in the *infer* task, regardless of whether the context story is shuffled or ordered.

## 5.2 INFLUENCE OF SPATIAL SIZE AND OBJECT NUMBER ON TASK DIFFICULTY.

The investigation based on four models including two thinking models (DeepSeek-R1, Doubao1.6-thinking-0715) and two non-thinking models (GPT-4o, Doubao1.5-pro-32k) for the influence of spatial size and object number on task difficulty, reveals that increased spatial size impairs model performance, whereas a greater number of objects improves it (Fig.3, 4).

When different values are assigned to the variable *Global Size*, while keeping other parameters consistent, the comprehensive performance of thinking models is significantly superior to that of non-thinking models. Additionally, as global size increases, the performance of both models shows a downward trend (Fig.3). Fig.3 b1-b3 respectively illustrate the accuracy of different models under two task types (infer and recall) and two context story conditions (shuffled and ordered), when the global size is set to 9, 11, and 13. The results clearly demonstrate that the model performance in the infer task type is far lower than that in the recall task type. In particular, the non-thinking model

Table 2: Model performance across four conditions in 3D space (%). SpintBench refers to the infer-shuffled column. Δ denotes the difference between shuffled and ordered scores (shuffled - ordered).

| Model | Infer | | | Recall | | | avg_score |
|---|---|---|---|---|---|---|---|
| | shuffled | ordered | Δ | shuffled | ordered | Δ | |
| Gemini-2.5-Pro | 48.00 | 55.00 | -7.00 | 69.00 | 67.00 | 2.00 | 59.75 |
| Gemini-2.5-Flash | 43.00 | 70.00 | -27.00 | 95.00 | 93.00 | 2.00 | 75.25 |
| OpenAI-o3-high.code | 34.00 | 64.00 | -30.00 | 95.00 | 92.00 | 3.00 | 71.25 |
| QwenAPI-3-235b-a22b-thinking | 21.00 | 60.00 | -39.00 | 96.00 | 94.00 | 2.00 | 67.75 |
| DeepSeekAPI-R1-0528 | 21.00 | 33.00 | -12.00 | 97.00 | 99.00 | -2.00 | 62.50 |
| Doubao-1.6-thinking.0715 | 15.00 | 25.00 | -10.00 | 98.00 | 97.00 | 1.00 | 58.75 |
| Claude-4-Opus-thinking | 13.00 | 58.00 | -45.00 | 90.00 | 98.00 | -8.00 | 64.75 |
| OpenAI-o4-mini.high.0416.code | 12.00 | 32.00 | -20.00 | 95.00 | 91.00 | 4.00 | 57.50 |
| Claude-4-Sonnet-thinking | 7.00 | 40.00 | -33.00 | 94.00 | 97.00 | -3.00 | 59.50 |
| QwenAPI-3-32b | 4.00 | 5.00 | -1.00 | 68.00 | 58.00 | 10.00 | 33.75 |
| QwenAPI-max-latest | 3.00 | 1.00 | 2.00 | 52.00 | 51.00 | 1.00 | 26.75 |
| GPT4o-1120 | 1.00 | 1.00 | 0.00 | 57.00 | 63.00 | -6.00 | 30.50 |
| Doubao1.5-pro-32k.250115 | 1.00 | 2.00 | -1.00 | 66.00 | 67.00 | -1.00 | 34.00 |
| Kimi-K2.volc.foreval | 1.00 | 0.00 | 1.00 | 94.00 | 84.00 | 10.00 | 44.75 |
| DeepSeekAPI-V3-0324 | 1.00 | 0.00 | 1.00 | 63.00 | 68.00 | -5.00 | 33.00 |
| OpenAI-gpt4.1-0414 | 0.00 | 1.00 | -1.00 | 84.00 | 85.00 | -1.00 | 42.50 |
| QwenAPI-qwq-plus.latest | 0.00 | 5.00 | -5.00 | 88.00 | 86.00 | 2.00 | 44.75 |

scores close to zero under the infer condition, while achieving relatively high scores under the recall condition. The shuffled condition is significantly more difficult than the ordered condition, which is consistent with the findings above.

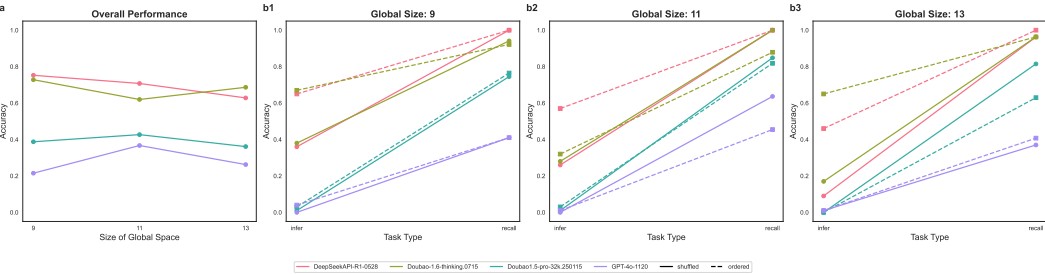

Figure 3: Influence of the size of *Global Space* on task difficulty. The experiment evaluates the performance of thinking and non-thinking models with varying global sizes (9, 11, and 13) across two task types (infer and recall) and two context conditions (shuffled and ordered).

However, when the number of objects in the space is increased while other parameters are kept invariant, model performance improves significantly (Fig.4 b1-b3). Furthermore, when the number of objects exceeds a certain threshold, even non-thinking models can achieve a substantial performance boost under the infer condition. These findings indicate that the density of objects in the space has a significant impact on task difficulty. Within a reasonable range, when two conditions are met—i.e., local spaces contain objects, and the combination of different local spaces results in a unique global space—a higher density of objects in the space corresponds to relatively lower task difficulty. This also implies that richer cues for spatial integration enable the model to more easily accomplish the integration and reasoning process from local spaces to the global space.

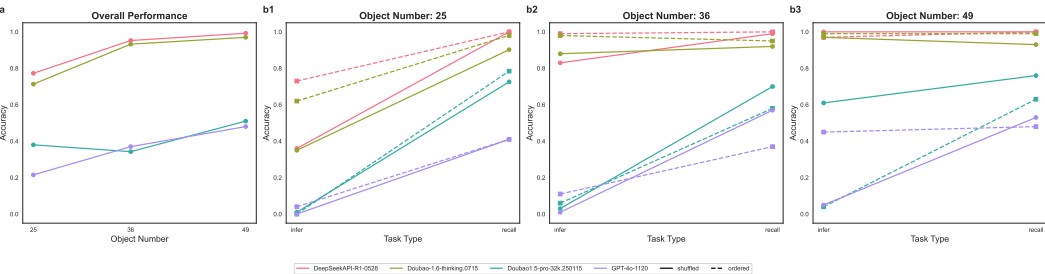

Figure 4: Influence of the size of *Object Number* on task difficulty. Increasing the number of objects in the space (25, 36, and 49), while keeping other parameters constant, leads to significant improvements in model performance

## 5.3 PROMPTING EXPERIMENTS

The prompting experiments also rely on four models as well, namely, two thinking models (DeepSeek-R1, Doubao1.6-thinking-0715) and two non-thinking models (GPT-4o, Doubao1.5-pro-32k). Results demonstrate that CoT One-Shot prompting is effective for most of these models—with a notable improvement observed in DeepSeek-R1-0528 (Table.3), whose performance is enhanced by five percent relative increase compared to the baseline. Overall, few-shot prompting facilitates the spatial reasoning capabilities of LLMs, and for certain LLMs, this facilitative effect could be further strengthened by integrating CoT or ReAct prompting strategies.

Table 3: Accuracy of Four Thinking or Non-thinking LLMs on Different PE Methods (%).

| PE method | DeepSeek-R1 | Doubao-1.6-thinking | Doubao1.5-pro-32k | GPT4o |
|---|---|---|---|---|
| Baseline (Zero-Shot) | 33 | 33 | 1 | 1 |
| ReAct One-Shot | 34↑ | 34↑ | 4↑ | 2↑ |
| CoT One-Shot | 38↑ | 34↑ | 4↑ | 3↑ |
| One-Shot | 35↑ | 36↑ | 3↑ | 1 |
| CoT Zero-Shot | 37↑ | 35↑ | 1 | 0 |

## 5.4 ERROR ANALYSIS

To further investigate the error mechanisms of LLMs, a qualitative analysis was conducted, encompassing both thinking and non-thinking models. The results indicated that under the *infer-shuffled* condition, non-thinking models exhibit a propensity for adopting heuristic shortcuts: they directly presuppose that the numbering of local grids conforms to a sequential arrangement, and subsequently process offsets and integrate local grids into a global space based on this unsubstantiated premise. This flawed methodological framework ultimately culminates in error generation, as corroborated by the performance of GPT-4o.

In contrast, thinking models (e.g., DeepSeek-R1-0528) eschew such shortcut-driven assumptions. Instead, they engage in reasoning anchored in shared object cues, following a structured procedural workflow: first, localizing the top-left corners of individual local spaces; second, defining coordinate variables, constructing mathematical equations leveraging the inter-variable relationships, and solving these equations to rigorously deduce the global spatial coordinates of each object; finally, calculating the Euclidean distance between the two target objects.

This observed behavioral discrepancy may bear a strong analogy to the dual cognitive systems (System 1 and System 2), with each model category aligning with one of these systems. Specifically, System 1—functionally analogous to non-thinking models—prioritizes rapid problem-solving relying on heuristic intuition; in contrast, System 2—corresponding to thinking models—depends on deliberate, iterative thinking and systematic reasoning to address tasks in a methodologically rigorous manner.

For thinking models, while they can solve a larger proportion of problems, the *infer-shuffled* tasks they correctly complete are typically those involving adjacent local spaces—requiring only one spatial

integration inference step. However, when the two target objects to be inferred are situated in local spaces that demand two or more spatial integration steps, the models exhibit a notable decline in performance. This finding highlights a persistent limitation of current reasoning models: they remain inadequate at tackling complex tasks that necessitate longer reasoning chains.

## 6 CONCLUSION

This study proposes a spatial integration framework applicable to both 2D and 3D spaces, to evaluate the ability of models to perform complex spatial reasoning. We additionally explores how the values of space-related parameters influence task difficulty and model performance. The results indicate that **SpintBench**—our spatial integration reasoning benchmark—poses substantial challenges for SOTA thinking models, while it is extremely more difficult for non-thinking models, with their scores approaching zero. Within a reasonable range, sparser objects in the space (i.e., lower density) correspond to greater reasoning difficulty. Prompting experiments demonstrate that combining few-shot learning with CoT prompting can improve model performance to a certain extent, yet this improvement fails to reach the desired level. These findings suggest that the spatial reasoning capabilities of current SOTA models still require further enhancement.

## 7 LIMITATIONS

This study has the following limitations and directions for future exploration. Since the data is automatically generated via predefined parameters and rules, the solvability and uniqueness of the questions require post-hoc manual verification. Although no cases failing to meet the criteria were identified during the quality inspection process, strict theory-driven proof is still advisable to confirm the uniqueness of solutions derived from inferring global space from local spaces.

Furthermore, when spatial integration is extended to 3D space, the number of local spaces within the global space increases with the cube of spatial dimensions. This results in relatively long context stories, posing challenges to the models' long-text processing capabilities as well. However, this also indicates that **SpintBench** has the potential to be expanded into a long-text evaluation benchmark for spatial reasoning, thereby providing some inspiration for future research.

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

# A APPENDIX

## A.1 PROMPTS USED IN EXPERIMENTS

---

**Baseline Prompt**

You are a reasoning expert. Please answer questions in strict accordance with the following format requirements:
1. First, present the complete calculation process, using LaTeX formulas to represent mathematical operations.
2. Keep the answer to two decimal places and do not include irrational numbers, square roots, or other such symbols.
3. Enclose the final answer with \boxed{final answer}.
{Question}

---

**One-Shot Prompt**

You are a reasoning expert. Please answer questions in strict accordance with the following format requirements:
1. First, present the complete calculation process, using LaTeX formulas to represent mathematical operations.
2. Keep the answer to two decimal places and do not include irrational numbers, square roots, or other such symbols.
3. Enclose the final answer with \boxed{final answer}.
<Example Start> Question:
Suppose there are three objects on a 3x3 2D plane, and each object is located in a small square cell. The length of each cell is 1 unit. Each time you can only see the plane of cell size 2x2. Now you see two 2x2 cells, and the relative coordinates of the objects inside are as follows: Local grid 1: Cat (local coordinates [1,1]), Dog (local coordinates [2,2]); Local grid 2: Dog (local coordinates [1,1]), Bird (local coordinates [1,2]). Please infer what the Euclidean distance is between Cat and Bird? Please keep two decimal places for the answer.
Answer:
Since Local Grid 1 and Local Grid 2 both contain "Dog", the two local grids are aligned and merged into a single global space with "Dog" as the reference. After the merging process, the coordinates of all objects are as follows: Cat [1,1], Dog [2,2], and Bird [3,2]. Therefore, the Euclidean distance between Cat and Bird is calculated as sqrt((1-3)² + (1-2)²) = 2.24.
</Example End>
{Question}

---

**CoT Zero-Shot Prompt**

You are a reasoning expert. Please answer questions in strict accordance with the following format requirements:
1. First, present the complete calculation process, using LaTeX formulas to represent mathematical operations.
2. Keep the answer to two decimal places and do not include irrational numbers, square roots, or other such symbols.
3. Enclose the final answer with \boxed{final answer}.
**Let's think step by step.**
{Question}

---

## A.2 PROMPT USED IN EXPERIMENTS

594
595
596
597
598
599
600
601
602
603
604
605
606
607
608
609
610
611
612
613
614
615
616
617
618
619
620
621
622
623
624
625
626
627
628
629
630
631
632
633
634
635
636
637
638
639
640
641
642
643
644
645
646
647

**CoT One-Shot Prompt**

You are a reasoning expert. Please answer questions in strict accordance with the following format requirements:

1. First, present the complete calculation process, using LaTeX formulas to represent mathematical operations.

2. Keep the answer to two decimal places and do not include irrational numbers, square roots, or other such symbols.

3. Enclose the final answer with \boxed{final answer}.

**Let's think step by step.**

<Example Start> Question:

Suppose there are three objects on a 3x3 2D plane, and each object is located in a small square cell. The length of each cell is 1 unit. Each time you can only see the plane of cell size 2x2. Now you see two 2x2 cells, and the relative coordinates of the objects inside are as follows: Local grid 1: Cat (local coordinates [1,1]), Dog (local coordinates [2,2]); Local grid 2: Dog (local coordinates [1,1]), Bird (local coordinates [1,2]). **Please infer what the Euclidean distance is between Cat and Bird? Please keep two decimal places for the answer.**

Answer:

Since Local Grid 1 and Local Grid 2 both contain "Dog", the two local grids are aligned and merged into a single global space with "Dog" as the reference. After the merging process, the coordinates of all objects are as follows: Cat [1,1], Dog [2,2], and Bird [3,2]. Therefore, the Euclidean distance between Cat and Bird is calculated as sqrt((1-3)² + (1-2)²) = 2.24.

</Example End>

{Question}

> **ReAct One-Shot Prompt**
>
> You are a reasoning expert. Please answer questions in strict accordance with the following format requirements:
> 1. First, present the complete calculation process, using LaTeX formulas to represent mathematical operations.
> 2. Keep the answer to two decimal places and do not include irrational numbers, square roots, or other such symbols.
> 3. Enclose the final answer with \boxed{final answer}.
> **You are supposed to perform dynamic reasoning to create, maintain, and adjust plans for acting while also enabling interaction to external environments to incorporate additional information into the reasoning.**
> <Example Start> Question:
> Suppose there are three objects on a 3x3 2D plane, and each object is located in a small square cell. The length of each cell is 1 unit. Each time you can only see the plane of cell size 2x2. Now you see two 2x2 cells, and the relative coordinates of the objects inside are as follows: Local grid 1: Cat (local coordinates [1,1]), Dog (local coordinates [2,2]); Local grid 2: Dog (local coordinates [1,1]), Bird (local coordinates [1,2]). **Please infer what the Euclidean distance is between Cat and Bird? Please keep two decimal places for the answer.**
> Answer:
> **Thought 1**: Since Local Grid 1 and Local Grid 2 both contain "Dog", the two local grids can be aligned and merged into a single global space with "Dog" as the reference.
> **Act 1**: Merge two local grids into one global space with "Dog" as the reference.
> **Thought 2**: After the merging process, the coordinates of all objects are as follows: Cat [1,1], Dog [2,2], and Bird [3,2]. Therefore, the Euclidean distance between Cat and Bird could be calculated.
> **Act 2**: Calculate the Euclidean distance between Cat and Bird. sqrt((1-3)² + (1-2)²) = 2.24.
> **Thought 3**: The Euclidean distance between Cat and Bird is 2.24.
> **Act 3**:Finish.
> </Example End>
> {Question}

