# OpenReview forum: "SpintBench: Evaluating LLMs' Complex \\ Reasoning via Spatial Integration Challenges"
_ICLR.cc/2026/Conference — Submitted to ICLR 2026_

### Official Review · Reviewer_FNLT · 2025-10-16

**Soundness:** 3
**Presentation:** 3
**Contribution:** 2
**Rating:** 4
**Confidence:** 3

**Summary:**

SpintBench introduces a text-only benchmark for spatial integration reasoning in both 2D and 3D, where models reconstruct a global map from overlapping local grids and answer Euclidean distance queries between two objects. The construction is automated: a global m×m grid is populated with n objects, partitioned into k×k local grids with one-row/column overlaps; ordered or shuffled “context stories” list local coordinates. Results show strong difficulty and discriminative power—especially infer-shuffled—where even top “thinking” models perform modestly.

**Strengths:**

1. The local-to-global integration setup is well motivated and implemented with ordered/shuffled contexts
2. CoT/few-shot yields modest improvements, with clear prompt templates in the appendix.

**Weaknesses:**

1. The paper acknowledges a significant theoretical gap, as it does not provide a proof for the uniqueness or solvability of global reconstruction from local overlaps. This leaves open questions about the reliability and consistency of the model's output.
2.  The empirical validation is constrained by a limited inference test size, which consists of only 100 samples. This small scale may not be sufficient to generalize the model's performance and robustness across a wider range of data.
3. The reliance on distance-only queries may limit the ecological validity of the findings. This approach potentially underrepresents other critical spatial reasoning skills, such as orientation and relative positioning, which are integral to comprehensive spatial understanding.
4. The authors assert that the model is resistant to data contamination; however, this claim is not substantiated with empirical data leakage checks or a rigorous theoretical justification. The absence of this validation makes it difficult to assess the true robustness of the model against contaminated training data.
5. While the 3D study includes comparative results against other methods, it offers a limited analysis of key performance factors. Specifically, it lacks a deep dive into how the model's performance is affected by scaling, variations in context length, and overall robustness to different conditions.

**Questions:**

Could you broaden the task set to include relation classification, direction/orientation, connectivity/topology, and step-count diagnostics for multi-hop spatial integration

Would you add algorithmic baselines (graph/constraint solvers) to contextualize LLM performance and validate benchmark difficulty

For 3D, can you study context length systematically (grid count, overlap degree) and propose scalable summarization or memory mechanisms to mitigate long-text effects

---

### Official Review · Reviewer_AfBy · 2025-10-31

**Soundness:** 2
**Presentation:** 3
**Contribution:** 2
**Rating:** 4
**Confidence:** 4

**Summary:**

This paper proposes a benchmark construction framework for evaluating spatial reasoning in both 2D and 3D spaces. It designed rules to automatically generate spatial descriptions of local scenes with overlapping cues, as well as corresponding question-answer (QA) pairs. It shows SOTA LLMs still struggle in their proposed benchmark SpintBench.

**Strengths:**

1. This paper find a meaningful automatic generation method to create benchmark which measures a weakness of current reasoning models, which is interesting.

**Weaknesses:**

1. This paper is lack of description of the differences between this work and similar previous works, for example StepGame, although relevant works already mentioned in this paper, but they has a lot of similarity with this work which is ignored and didn't mention in this paper, e.g. some of them also are multi-hop spatial reasoning using so called transitive inference. Therefore, I cannot confirm the novelty of this paper. Moreover, there are existing similar works which this paper didn't discuss, e.g. [1], they also proposed to use automatic generated benchmark for testing, and discussed scenarios of 2D and 3D, 'shuffled' and without shuffled. Can you tell the differences of your work with theirs?

2. Although this paper claimed its about spatial reasoning, but in my understanding, its more like symbolic reasoning based on spatial description. Unlike some spatial reasoning benchmarks which contains visual information requiring models to conduct 'real' spatial reasoning, the proposed benchmark require models has symbolic reasoning abilities, computation abilities, and memories.

3. Lack of human performance. Although a lot of models are compared and evaluated in this paper, this paper didn't discuss and cover the performance of human in their proposed task.

[1] Chain-of-Symbol Prompting for Spatial Reasoning in Large Language Models.

**Questions:**

Please refer to the weaknesses.

---

### Official Review · Reviewer_Qg5u · 2025-10-31

**Soundness:** 2
**Presentation:** 3
**Contribution:** 2
**Rating:** 4
**Confidence:** 3

**Summary:**

The paper proposes a new text-based benchmark for spatial relations, called SpintBench. This benchmark is designed to evaluate a model’s ability to infer global spatial information from given local information. The authors evaluate 17 large language models (LLMs), including both language-only and multimodal models. The results show that while these models can handle recall-based, within the same grid, inferences in finding distance task, LLMs struggle to combine local knowledge and derive a coherent global understanding of spatial relations.
The paper also presents an investigation into the effects of global size and object count on task accuracy. Finally, the authors perform an error analysis and find that non-thinking models often assume local grids are close to each other and ignore common objects shared between grids. In contrast, thinking models avoid such shortcut assumptions and perform more deliberate reasoning, leading to better performance.

**Strengths:**

- The proposed benchmark effectively evaluates LLMs’ ability to perform mental reasoning and reconstruct global spatial relations from local information. This addresses an important problem in spatial reasoning, enabling models to infer the structure of a global environment based on partial, local cues.

- The paper provides detailed evaluations and ablation studies on their dataset, covering three key conditions: order, shuffle, and recall-infer. Order-shuffle refers to ordered of grid presented in context. Recall-infer refers to whether the question can be answered within grid, or required merging.

- Paper evaluate both 2D and 3D settings for this problem. The results is similar between two domains that the model fail to reconstructthe global representation based on given local cues.

- The paper also includes a detailed failure analysis, revealing how non-thinking and thinking models differ in their reasoning strategies when solving spatial relation tasks.

- Well written paper, and provide illsutration to help explain the evaluated task.

**Weaknesses:**

- Figure 3, which illustrates model performance across different global sizes for inference and recall questions, may cause confusion, as it can be misinterpreted to suggest that one result represents an improvement over the other. Consider revising or clarifying this visualization to better distinguish the two.

- Although an error analysis is provided, it lacks sufficient detail. Including 2–3 sentences explaining how the analysis was conducted would improve clarity and help ensure the reproducibility of the findings.

- While the task design is interesting, the current evaluation (based on computing Euclidean distance) may be somewhat impractical. A model might still compute the correct distance despite having an incorrect understanding of the layout or swapped object positions. Extending the benchmark to include more practical downstream tasks such as relation identification, layout reconstruction, or compositional reasoning would make the evaluation more meaningful.

- The error analysis highlights that models struggle with multi-step reasoning; however, it lacks deeper discussion or examples demonstrating the specific types of reasoning failures. Expanding this section to categorize and illustrate the observed errors would strengthen the paper’s insights into model behavior.

- No initial prompt engineering or methodological approach is proposed to address the shortcomings of LLMs in this task, aside fromexcept utilizing CoT which should be considered as baseline.

**Questions:**

- Is there any difference in performance between models trained with language-only data and those trained with multimodal data when interacting with grids or combining local information?
- Is there a specific reason for selecting Euclidean distance as the downstream task over other potential tasks such as relation identification or layout reconstruction?

---

### Official Review · Reviewer_vakV · 2025-11-07

**Soundness:** 2
**Presentation:** 2
**Contribution:** 1
**Rating:** 2
**Confidence:** 4

**Summary:**

This paper introduces SpintBench, an automatically generated benchmark to evaluate spatial integration reasoning in both 2D and 3D contexts. It extends transitive inference from 1D to higher dimensions, requiring LLMs to synthesize local spatial cues into global configurations. The benchmark evaluates 17 models and explores effects of parameters, spatial density, and prompting methods (CoT, ReAct). The study concludes that even “thinking” models struggle, suggesting persistent limits in spatial reasoning.

**Strengths:**

1. Novel angle: Evaluating spatial integration reasoning in text-only LLMs is timely and relatively unexplored.

2. Automatic generation pipeline: Offers scalability and resistance to data contamination.

**Weaknesses:**

1. Misaligned problem definition (the teaser fails to represent the paper’s intent).

The teaser and introduction claim to study “spatial reasoning,” but the benchmark remains restricted to 2D distance estimation rather than reasoning about 3D spatial relations or physical layouts. Traditional spatial reasoning involves object localization and relational understanding in 3D (e.g., topology, containment, occlusion, perspective), not mere 2D metric inference. The conceptual framing thus misrepresents the task scope and inflates the claimed contribution.

2. Insufficient benchmark scale and unclear data composition.

The dataset is relatively small, and the paper lacks comprehensive statistics or visual summaries of the benchmark. Readers cannot assess the distribution of grid sizes, object counts, or overlap ratios. Without transparent visualization (e.g., histograms or embedding maps), the dataset’s diversity and representativeness remain uncertain.

3. Poor organization and weak narrative flow.

The paper’s exposition is disjointed and difficult to follow. Key concepts are introduced abruptly, and methodological details are scattered across sections without coherent logic. Many paragraphs mix motivation, implementation, and results. Substantial restructuring and language polishing are needed for clarity and readability.

4. Oversimplified evaluation dimension.

The current evaluation focuses only on a single scalar distance metric, which cannot sufficiently reflect a model’s true spatial reasoning ability. Spatial reasoning should involve qualitative relations (left/right/front/behind), multi-object configurations, or relational inference beyond metric computation. The current setting thus evaluates numerical estimation, not reasoning.

**Questions:**

I am curious about the true value of the dataset, as I strongly suspect that it may merely overfit to its own format.

If a base model (e.g., Qwen-VL) could be fine-tuned on this dataset and subsequently demonstrate performance gains on other benchmarks (such as VSI-Bench, OmniSpatial and SPACE), I would be much more inclined to recognize the dataset’s contribution.

---

### Meta-Review · Area_Chair_B4Z6 · 2025-12-31

**Summary:**

This paper was reviewed by four experts who all recommended rejection.  Reviewers noted issues with the paper's writing, motivation, and weak experiments.  There was no rebuttal.

**Reviewer Concerns:**

N/A, no rebuttal

**Reviewer Scores:**

As there was no rebuttal so reviewers would likely have either maintained or lowered their scores.

---

### Decision · Program_Chairs · 2026-01-26

Reject